# Influence of Al-O and Al-C Clusters on Defects in Graphene Nanosheets Derived from Coal-Tar Pitch via Al_4_C_3_ Precursor

**DOI:** 10.3390/ma15207312

**Published:** 2022-10-19

**Authors:** Peng Lin, Yinggan Zhang, Zhou Cui, Rui Xiong, Cuilian Wen, Bo Wu, Qilang Lin, Baisheng Sa

**Affiliations:** 1Key Laboratory of Eco-Materials Advanced Technology, College of Materials Science and Engineering, Fuzhou University, Fuzhou 350108, China; 2Fujian Provincial Key Laboratory of Theoretical and Computational Chemistry, College of Materials, Xiamen University, Xiamen 361005, China

**Keywords:** graphene nanosheets, X-ray photoelectron spectroscopy, first-principles calculations, defects

## Abstract

By treating Al_4_C_3_ as the precursor and growth environment, graphene nanosheets (GNs) can efficiently be derived from coal-tar pitch, which has the advantages of simple preparation process, high product quality, green environmental protection, low equipment requirements and low preparation cost. However, the defects in the prepared GNs have not been well understood. In order to optimize the preparation process, based on density functional theory calculations, the influence mechanism of Al-O and Al-C clusters on defects in GNs derived from coal-tar pitch via Al_4_C_3_ precursor has been systematically investigated. With minute quantities of oxygen-containing defects, Al-O and Al-C clusters have been realized in the prepared GNs from X-ray photoelectron spectroscopy analysis. Therefore, the influences of Al-O and Al-C clusters on graphene with vacancy defects and oxygen-containing defects are systematically explored from theoretical energy, electron localization function and charge transfer analysis. It is noted that the remaining Al-O and Al-C clusters in GNs are inevitably from the thermodynamics point of view. On the other hand, the existence of defects is beneficial for the further adsorption of Al-O and Al-C clusters in GNs.

## 1. Introduction

In 2004, Geim and Novoselov obtained graphene for the first time by mechanical exfoliation [1]. Graphene shows excellent physical and chemical properties such as high electron mobility, high thermal conductivity and large specific surface area [2,3,4,5]. Therefore, graphene has been widely used in aerospace, solar cells, nanoelectronics, electronic devices and other fields [6]. It is noted that one of the key problems restricting the application of graphene is how to achieve large-scale, reproducible and low-cost preparation of high-quality crystalline graphene nanosheets (GNs) with regular structure, controllable thickness and controllable size [7,8]. At present, various ways have been proposed for the preparation of GNs, for instance, the micromechanical exfoliation method [1,9], the liquid phase exfoliation method [10,11], the solvothermal method [12,13], the supercritical fluid exfoliation method [14,15], the chemical exfoliation method [16,17], the electrochemical exfoliation method [18,19], the cutting carbon nanotube method [20,21], the chemical vapor deposition method [22,23], the epitaxial growth method [24,25], the in situ self-generated template method [26], the organic synthesis method [27,28] and the arc discharge method [29]. However, it is still an open challenge to develop environmentally friendly methods for GNs [30,31,32].

Coal-tar pitch is an abundant and cheap natural resource which has been proposed as the carbon source for the low-cost synthesis of graphene and related materials [33,34,35]. Using coal pitch and aluminum powder as raw materials, we have prepared GNs with aluminum carbide as the intermediate by high-temperature pyrolysis and the acid pickling method [36,37]. This method has the advantages of a simple preparation process, high product quality, green environmental protection, lower requirements for equipment and low preparation cost [36,37]. Nonetheless, we found that the prepared GNs still have a small amount of defects [36,37], which could play a critical role in the functionality and performance of graphene [38]. On the other hand, in the pyrolysis reaction, due to the presence of residual oxygen in the furnace, the aluminum carbide intermediate will be oxidized to Al-O clusters, which show strong adsorption properties and will be adsorbed on the defects of GNs [36,37]. In addition, as the most important intermediate in the pyrolysis reaction, Al_4_C_3_ could be decomposed into Al-C clusters and adsorbed on the defects of GNs [36,37]. It is worth noting that density functional theory (DFT) based simulation methods give us a powerful tool to unravel the electronic origin of defects in materials in atomically scale [39,40,41]. Therefore, it is of great interest and importance to understand the formation mechanism of the defects as well as the influence mechanism of Al-O and Al-C clusters in the obtained GNs derived from coal-tar pitch by using Al_4_C_3_ as an intermediate.

In this work, by combining the X-ray photoelectron spectroscopy measurements and density functional theory calculations, we systematically investigated the adsorption performance of Al-O and Al-C clusters on different defects in GNs. The adsorption properties of Al-O and Al-C clusters on nine defected graphene models as well as the perfect graphene have been studied by energy, electron localization function and charge transfer analysis. Our work provides theoretical basis and technical support for improving the preparation process and the quality of GNs from coal-tar pitch.

## 2. Experiment and Calculation Methods

### 2.1. Materials and Characterizations

In this paper, following the optimal graphene preparation process studied by our previous work [37], using coal tar and aluminum powder as raw materials, treating aluminum carbide as the intermediate and precursor, GNs were prepared by high-temperature pyrolysis reaction at 1500 °C and HCl acid washing via a four-step synthesis method. X-ray photoelectron spectroscopy (XPS, K-Alpha^+^) was used to detect the relative content and chemical valence state of each atom and functional group of the graphene nanosheets after acid washing. Monochromatic Al K*α* was used as the excitation source. The “Thermo Avantage 5.52” software was used for the XPS data deconvolutions.

### 2.2. Calculation Methods

DFT calculations using the projected augmented wave method implemented in the Vienna ab-initio simulation package (VASP) code were performed [42,43,44]. The generalized gradient approximation (GGA) Perdew–Burke–Ernzerhof (PBE) [45] is used for the exchange-correlation pseudopotentials. Plane-wave basis set was used with an energy cutoff of 500 eV. Both relaxation convergence for electrons and ions were 1 × 10^−4^ eV. We adopted graphene in the orthorhombic lattice with a 10 × 10 × 1 supercell including 100 C atoms for the defect models. A k-mesh of 1 × 1 × 1 was used for the sampling of the Brillouin zone. The van der Waals (vdW) interaction was considered at the DFT-D3 level as proposed by Grimme [46]. We used the ALKEMIE platform to deal with the calculation results [47].

## 3. Results and Discussion

The prepared GNs following the same synthesis condition from our previous work [37] were analyzed by X-ray photoelectron spectroscopy (XPS), as shown in Figure 1. Table 1 lists the atomic concentration and peak binding energy of our prepared GNs. Figure 1a shows the full XPS spectrum of GNs, where the peak positions of C1s, O1s and Al 2p are marked. Herein, the strength of C1s peak is much higher than the others, indicating the carbon bonding dominates the samples. The results agree well with our previous XRD, TEM and Raman results [37]. On the other hand, the O1s cannot be ignored in the full XPS spectrum. The result indicates that we have produced oxygen-containing defects in GNs. Moreover, since Al_4_C_3_ is treated as the precursor and growth environment for GNs, a weak Al2p peak is observed as well. Herein, the limited strength of the Al2p peak confirms that most of the Al_4_C_3_ precursor has been washed out in the acid washing process. Figure 1b further displays the C1s spectrum in detail. It can be seen that the C=C and C-C bonds dominate the C1s states. Moreover, the obtained GNs contain a small amount of C-O and Al-C bonds. Figure 1c illustrates the O1s spectrum in detail. It shows the existence of Al-O bonds and C-O bonds. The small amount of H-O-H peaks indicates the remaining adsorption H_2_O molecules in GNs after drying. Figure 1c is the Al2p spectrum in detail. The Al^3+^ peak refers to the remaining Al_4_C_3_ after the acid washing process and the Al^0^ peak refers to the existence of the Al-C or Al-O cluster in GNs. Combining the above spectrums, we confirmed the oxygen-containing defects and the Al-O and Al-C clusters in the obtained GNs. Therefore, we have built 9 different graphene defect models based on the XPS analysis for further DFT calculations. Figure 1e illustrates the optimized graphene defect models as well as the perfect graphene supercell. The calculated C-C bond length is 1.413 Å and agrees well with the well-known C-C bond length in graphene of 1.413 Å. To simplify, the abbreviation G refers to the perfect graphene. The abbreviations G-DV, G-SV and G-SW are short for double-vacancy defect, single-vacancy defect and Stone–Wales defect, respectively. G-O1 to G-O6 refer to the different possible oxygen-containing defect graphene models. The defect formation energies $E_{f}^{G}$ without any adsorptions for the defects models in Figure 1e were calculated according to [48]:(1)$$E_{f}^{G}\;=\;E_{\mathrm{defect}}\;-\;E_{G}\;+\;\mu_{C}\;-\;\mu_{O}$$
where *E*_defect_ is the total energy of graphene with defect, *E*_G_ denotes the total energy of perfect graphene and *μ*_C_ and *μ*_O_ represent the chemical potentials of missing C atom in graphene and additional O atom in oxygen gas, respectively. The calculated defect formation energies $E_{f}^{G}$ are listed in Table 2. The positive values indicate that the formation of the defects are endothermic processes.

To explore the effect of the Al-O cluster adsorption to graphene, the corresponding defect formation energy $E_{f}^{\mathrm{Al}-O}$ was calculated according to [48]:(2)$$E_{f}^{\mathrm{Al}-O}\;=\;E_{\mathrm{defect}}^{\mathrm{Al}-O}\;-\;E_{G}^{\mathrm{Al}-O}\;+\;\mu_{C}\;-\;\mu_{O}$$
where $E_{\mathrm{defect}}^{\mathrm{Al}-O}$ is the total energy of graphene with the defect and adsorption of the Al-O cluster and $E_{G}^{\mathrm{Al}-O}$ demotes the total energy of the perfect graphene with the adsorption of the Al-O cluster. The corresponding adsorption structures are illustrated in Figure 2. The calculated defect formation energies $E_{f}^{\mathrm{Al}-O}$ are listed in Table 2. The change of defect formation energies $\Delta E_{f}^{\mathrm{Al}-O}\;=\;E_{f}^{\mathrm{Al}-O}\;-\;E_{f}^{G}$ after the adsorption of the Al-O cluster are listed in Table 2 as well. It can be seen from Table 2 that the defect formation energy $E_{f}^{\mathrm{Al}-O}$ of all the defect models decreased after the adsorption of the Al-O cluster, which indicates that the adsorption of the Al-O cluster makes it easier for the graphene to form defects. The decreasing values from low to high were the G-O5, G-SW, G-O4, G-O1, G-O3, G-O2, G-SV, G-DV and G-O6 models. In addition, the change of the defect formation energy $\Delta E_{f}^{\mathrm{Al}-O}$ of the G-O6 model is significantly higher than that of other models, indicating that such an oxygen-containing defect is more significantly affected by the presence of the Al-O cluster.

A deeper understanding of the influence of the Al-O cluster on graphene with different defects can be gained by analyzing the electron localization functions (ELF) [49,50] and charge transfer properties [51]. Figure 3a shows the ELF plots in the (010) plane for different graphene defect models with the adsorption of the Al-O cluster. It shows a strong electron localization feature between the C-C bonds in graphene. For the G, G-SW, G-O1, G-O4 and G-O5 models, notably delocalized electrons with high ELF values close to 1 are observed around the Al-O clusters in the region around the Al atoms, which correspond to the Al^0^ states in the XPS spectrum and relate to the remaining unpaired electrons from Al. On the other hand, for the G-DV, G-SV, G-O2, G-O3 and G-O6 models, the delocalized electrons have disappeared, since the Al atom is strongly bonded to the graphene models. Figure 3b illustrates the isosurfaces of charge density difference $\rho_{\mathrm{diff}}^{\mathrm{Al}-O}$ for different graphene defect models after the adsorption of the Al-O cluster according to [52]:(3)$$\rho_{\mathrm{diff}}^{\mathrm{Al}-O}\;=\;\rho_{\mathrm{defect}+\mathrm{Al}-O}^{}\;-\;(\rho_{\mathrm{defect}}^{}\;+\;\rho_{\mathrm{Al}-O}^{})$$
where $\rho_{\mathrm{defect}+\mathrm{Al}-O}^{}$, $\rho_{\mathrm{defect}}^{}$ and $\rho_{\mathrm{Al}-O}^{}$ denote the charge densities of defect graphene with adsorption of Al-O cluster, defect graphene without adsorption of Al-O cluster and freely Al-O cluster, respectively. The cyan and yellow isosurfaces correspond to the loss and gain of electrons. Different charge transfer behaviors can be vividly shown from the 3D isosurface plots. Generally speaking, the more remarkable charge transfer corresponds to stronger interactions. Obviously, charge transfers between the Al-O cluster and graphene are observed for the G-DV, G-SV, G-O2, G-O3 and G-O6 models, indicating a strong interaction between the adsorbed Al-O cluster and the corresponding defects.

To quantitatively analyze the charge transfer between the adsorbed Al-O cluster and the graphene, we calculated the electron charge transfer amount $e_{\mathrm{trans}}^{\mathrm{Al}-O}$ according to [52]:(4)$$e_{\mathrm{trans}}^{\mathrm{Al}-O}\;=\;e_{\mathrm{free}}^{\mathrm{Al}-O}\;-\;e_{\mathrm{ad}}^{\mathrm{Al}-O}$$
where $e_{\mathrm{free}}^{\mathrm{Al}-O}$ and $e_{\mathrm{ad}}^{\mathrm{Al}-O}$ denote the summarized Bader charge [53] of the Al-O cluster part for the defect graphene models before and after adsorption, respectively. Figure 4a summarizes the calculated $e_{\mathrm{trans}}^{\mathrm{Al}-O}$. Herein, the positive and negative values of the electron charge transfer amount $e_{\mathrm{trans}}^{\mathrm{Al}-O}$ indicate the loss and gain of electrons of the Al-O cluster after adsorption, respectively. It shows that the charge transfer between perfect graphene without any defect and the Al-O cluster is only 0.02 *e*, indicating very weak interaction between them. The existence of all the defects will increase the charge transfer between the adsorbed Al-O cluster and graphene. The positive $e_{\mathrm{trans}}^{\mathrm{Al}-O}$ values around 1 for the G-DV, G-SV, G-O2, G-O3 and G-O6 models indicate obviously charge transfer from the adsorbed Al-O cluster to graphene with corresponding defects. Moreover, the negative $e_{\mathrm{trans}}^{\mathrm{Al}-O}$ values for the G-SW, G-O1, G-O4 and G-O5 models represent the charge accumulation around the adsorbed Al-O cluster for these cases, which agrees well with the previously delocalized states from the ELF analysis. To further quantitatively analyze the interaction between the adsorbed Al-O cluster and the graphene defect, we calculated the adsorption binding energy of the Al-O cluster $E_{b}^{\mathrm{Al}-O}$ according to [52]:(5)$$E_{b}^{\mathrm{Al}-O}\;=\;E_{\mathrm{defect}+\mathrm{Al}-O}^{}\;-\;(E_{\mathrm{defect}}^{}\;+\;E_{\mathrm{Al}-O}^{})$$

Figure 4b shows the calculated $-E_{b}^{\mathrm{Al}-O}$. Herein, the positive values of $-E_{b}^{\mathrm{Al}-O}$ correspond to the thermopositive reactions. The small positive value of $-E_{b}^{\mathrm{Al}-O}$ = 0.47 eV for the perfect graphene without any defect indicates that the adsorption of the Al-O cluster is an exothermic process. Therefore, the remaining Al-O cluster in our GN samples is inevitable from the thermodynamics point of view. On the other hand, more positive values of $-E_{b}^{\mathrm{Al}-O}$ for graphene with defects demonstrate the fact that the existence of any defects is beneficial for the further adsorption of the Al-O cluster.

To explore the effect of the Al-C cluster adsorption to the graphene, the corresponding defect formation energy $E_{f}^{\mathrm{Al}-C}$ was calculated according to [52]:(6)$$E_{f}^{\mathrm{Al}-C}\;=\;E_{\mathrm{defect}}^{\mathrm{Al}-C}\;-\;E_{G}^{\mathrm{Al}-C}\;+\;\mu_{C}\;-\;\mu_{O}$$
where $E_{\mathrm{defect}}^{\mathrm{Al}-C}$ is the total energy of graphene with the defect and the adsorption of Al-C cluster and $E_{G}^{\mathrm{Al}-C}$ demotes the total energy of the perfect graphene with the adsorption of Al-C cluster. The corresponding adsorption structures are illustrated in Figure 5. The calculated defect formation energies $E_{f}^{\mathrm{Al}-C}$ are listed in Table 2. The change of defect formation energies $\Delta E_{f}^{\mathrm{Al}-C}\;=\;E_{f}^{\mathrm{Al}-C}\;-\;E_{f}^{G}$ after the adsorption of the Al-C cluster are listed in Table 2 as well. It can be seen from Table 2 that after the adsorption of the Al-C cluster, the defect formation energy $E_{f}^{\mathrm{Al}-C}$ for most of the graphene defect models decrease. The decreasing values from low to high are the G-O4, G-O3, G-SW, G-SV, G-O2, G-O6 and G-DV models. These cases are similar to the Al-O cluster adsorption models. However, the defect formation energy $E_{f}^{\mathrm{Al}-C}$ for the G-O1 and G-O5 models increase after the adsorption of the Al-C cluster, indicating that the formation of some oxygen-containing defects can be reduced after the adsorption of the Al-C cluster.

Further analysis of the ELF and charge transfer properties is presented to understand the influence of the Al-C cluster on graphene with different defects in the following. Figure 6a shows the ELF plots in the (010) plane for the different graphene defect models with the adsorption of the Al-C cluster. It is similar to the Al-O cluster cases, notably the delocalized electrons with high ELF values close to 1 are observed around the Al-C clusters in the region around the Al atoms for the G, G-SW, G-O1, G-O4 and G-O5 models. At the same time, the delocalized electrons disappear for the G-DV, G-SV, G-O2, G-O3 and G-O6 models. The results indicate that the influence of the Al-C cluster is similar to the Al-O cluster. Figure 6b represents the isosurfaces of the charge density difference $\rho_{\mathrm{diff}}^{\mathrm{Al}-C}$ for different graphene defect models after the adsorption of the Al-C cluster according to [52]:(7)$$\rho_{\mathrm{diff}}^{\mathrm{Al}-C}\;=\;\rho_{\mathrm{defect}+\mathrm{Al}-C}^{}\;-\;(\rho_{\mathrm{defect}}^{}\;+\;\rho_{\mathrm{Al}-C}^{})$$
where $\rho_{\mathrm{defect}+\mathrm{Al}-C}^{}$, $\rho_{\mathrm{defect}}^{}$ and $\rho_{\mathrm{Al}-C}^{}$ denote the charge densities of defect graphene with the adsorption of the Al-C cluster, the defect graphene without the adsorption of the Al-C cluster and the free Al-C cluster, respectively. It is similar to the Al-O cluster cases as well, as obvious charge transfers between the Al-C cluster and graphene are observed for the G-DV, G-SV, G-O2, G-O3 and G-O6 models, indicating strong interaction between the adsorbed Al-C cluster and the corresponding defects.

The electron charge transfer amount $e_{\mathrm{trans}}^{\mathrm{Al}-C}$ is further calculated to quantitatively analyze the charge transfer between the adsorbed Al-C cluster and the graphene according to [52]:(8)$$e_{\mathrm{trans}}^{\mathrm{Al}-C}\;=\;e_{\mathrm{free}}^{\mathrm{Al}-C}\;-\;e_{\mathrm{ad}}^{\mathrm{Al}-C}$$
where $e_{\mathrm{free}}^{\mathrm{Al}-C}$ and $e_{\mathrm{ad}}^{\mathrm{Al}-C}$ denote the summarized Bader charge of the Al-C cluster part for the defect graphene models before and after adsorption, respectively. The calculated $e_{\mathrm{trans}}^{\mathrm{Al}-C}$ is illustrated in Figure 7a, where the positive values of the electron charge transfer amount $e_{\mathrm{trans}}^{\mathrm{Al}-C}$ for all the graphene defect models indicate the loss of electrons of the Al-C cluster after adsorption. The charge transfer between the perfect graphene without any defect and the Al-C cluster is 0.1 *e*, indicating very weak interaction similar to the Al-O cluster and the perfect graphene. In addition, the large positive $e_{\mathrm{trans}}^{\mathrm{Al}-O}$ values greater than 1 for the G-DV, G-SV, G-O2, G-O3 and G-O6 models indicate an obvious charge transfer from the adsorbed Al-C cluster to the graphene with corresponding defects, which is similar to the Al-O cluster cases as well. However, for the G-SW, G-O1, G-O4 and G-O5 models, positive $e_{\mathrm{trans}}^{\mathrm{Al}-O}$ values smaller than 1 are obtained, which differs from the negative values for the Al-O cluster cases. In spite of this, the charge transfer from the Al-C cluster to corresponding graphene defect models is limited. Therefore, the delocalized ELF states can be found for the G-SW, G-O1, G-O4 and G-O5 models. To further quantitatively analyze the interaction between the adsorbed Al-C cluster and the graphene defect, we calculated the adsorption binding energy of the Al-C cluster $E_{b}^{\mathrm{Al}-C}$ according to [52]:(9)$$E_{b}^{\mathrm{Al}-C}\;=\;E_{\mathrm{defect}+\mathrm{Al}-C}^{}\;-\;(E_{\mathrm{defect}}^{}\;+\;E_{\mathrm{Al}-C}^{})$$

Figure 7b shows the calculated $-E_{b}^{\mathrm{Al}-C}$. It is similar to the Al-O cluster cases, where the positive values of $-E_{b}^{\mathrm{Al}-C}$ correspond to the thermopositive reactions. The positive value of $-E_{b}^{\mathrm{Al}-C}$ = 0.73 eV for the perfect graphene without any defect indicates that the adsorption of the Al-C cluster is an exothermic process, and the interaction between the Al-C cluster and perfect graphene is greater than that of the Al-O cluster. Therefore, the remaining Al-C cluster in the GN samples is inevitable from the thermodynamics point of view. Moreover, except for the G-O1 and G-O5 models, more positive values of $-E_{b}^{\mathrm{Al}-O}$ for the graphene with defects demonstrates the fact that the existence of the other defects is beneficial for the further adsorption of the Al-O cluster. 

## 4. Conclusions

To conclude, by combining the X-ray photoelectron spectroscopy measurements and density functional theory calculations, we have systematically investigated the adsorption performance of the Al-O and Al-C clusters on perfect graphene and graphene with a double-vacancy defect, single-vacancy defect, Stone–Wales defect and different oxygen-containing defects. The minute quantities of oxygen-containing defects in Al-O and Al-C clusters have been realized in the prepared GNs from the additional XPS analysis. The adsorption properties of the Al-O and Al-C clusters have been revealed from the DFT calculations. The positive values of the defect formation energy indicate that the formation of the defects in graphene are endothermic processes, and the presence of Al-O and Al-C clusters makes it easier for the graphene to form most types of defects. Furthermore, the remaining Al-O and Al-C clusters in the GNs are inevitable from the thermodynamics point of view. On the other hand, the existence of defects is beneficial for the further adsorption of Al-O and Al-C clusters in the GNs.

## Figures and Tables

**Figure 1 materials-15-07312-f001:**
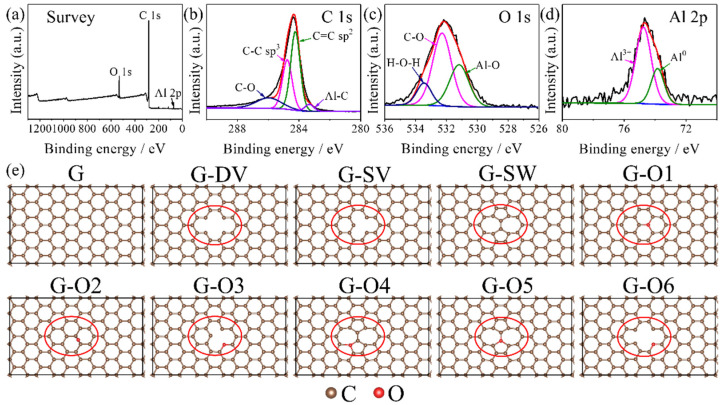
(**a**) Full spectrum, (**b**) C1s, (**c**) O1s and (**d**) Al2p XPS analysis of GNs. (**e**) The proposed graphene defect models, the red circles highlight the defects in graphene.

**Figure 2 materials-15-07312-f002:**
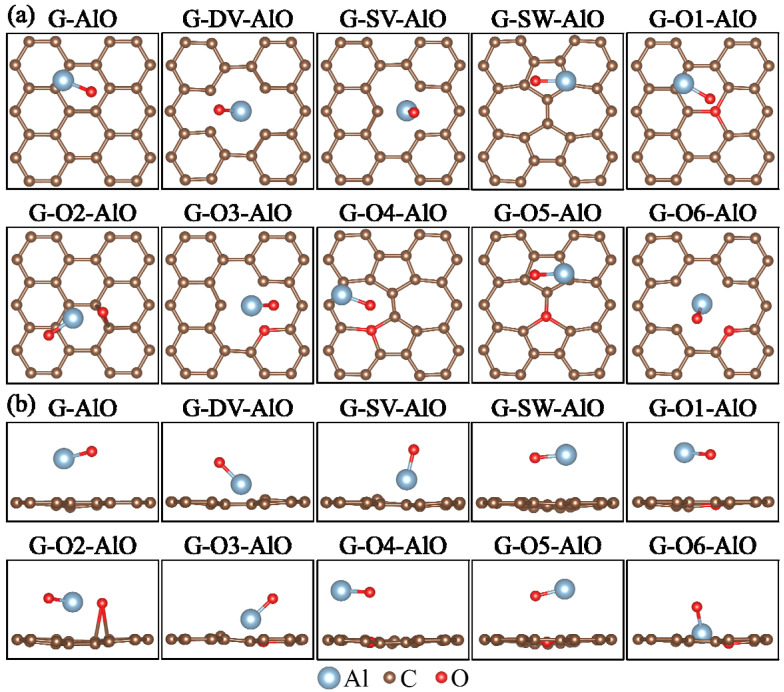
(**a**) Top and (**b**) side views of Al-O cluster adsorption structures for graphene defect models.

**Figure 3 materials-15-07312-f003:**
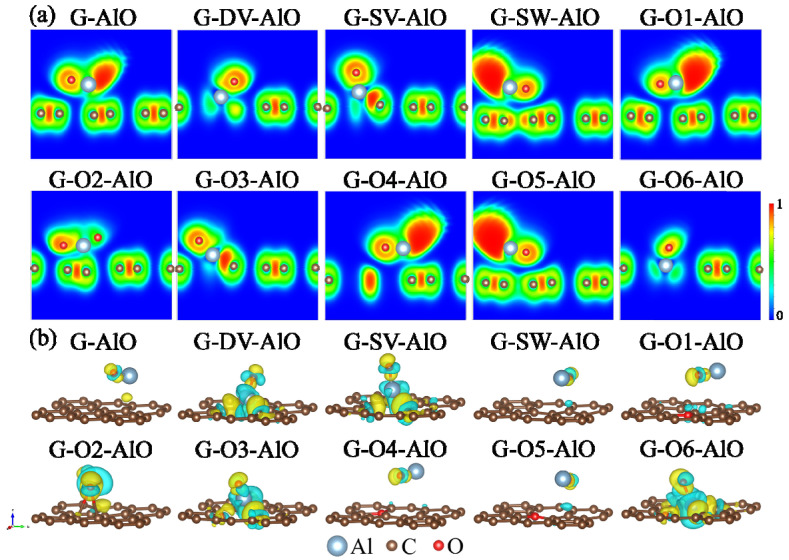
(**a**) The 2D plots of ELF in the (010) plane and (**b**) 3D plots of charge density difference of Al-O cluster adsorption structures for graphene defect models. The loss of electrons is indicated in cyan and the gain of electrons is indicated in yellow with an isosurface value of 0.005 e/au^3^.

**Figure 4 materials-15-07312-f004:**
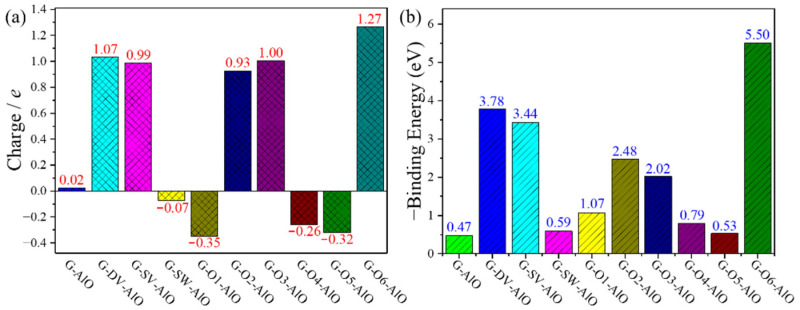
(**a**) Charge transfer amount and (**b**) adsorption binding energy of Al-O cluster adsorption structures for graphene defect models.

**Figure 5 materials-15-07312-f005:**
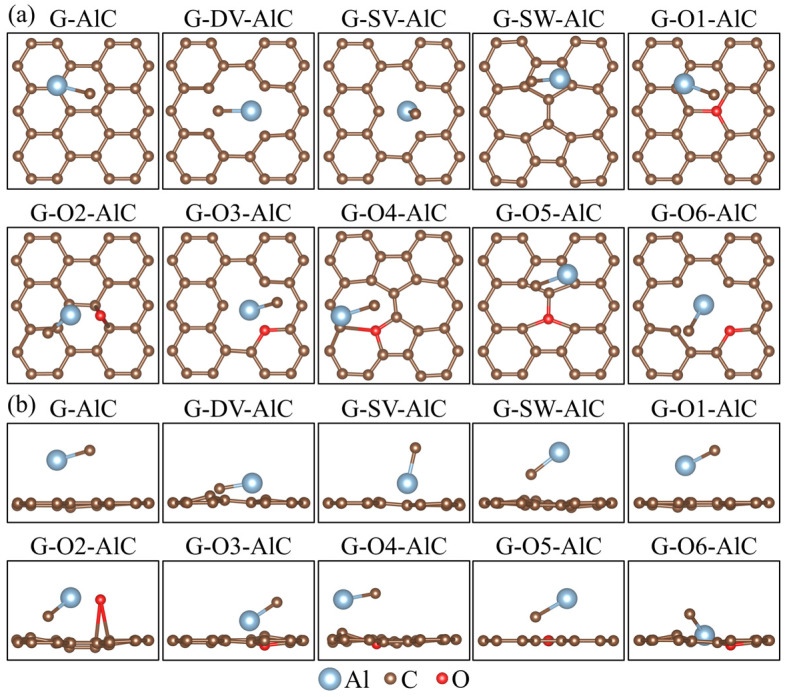
(**a**) Top and (**b**) side views of Al-C cluster adsorption structures for graphene defect models.

**Figure 6 materials-15-07312-f006:**
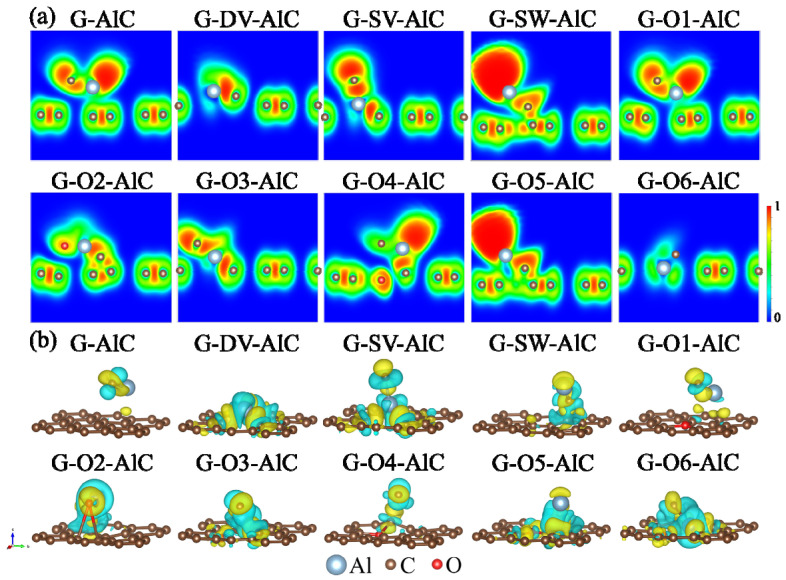
(**a**) The 2D plots of ELF in the (010) plane and (**b**) 3D plots of charge density difference of Al-C cluster adsorption structures for graphene defect models. The loss of electrons is indicated in cyan and the gain of electrons is indicated in yellow with an isosurface value of 0.005 e/au^3^.

**Figure 7 materials-15-07312-f007:**
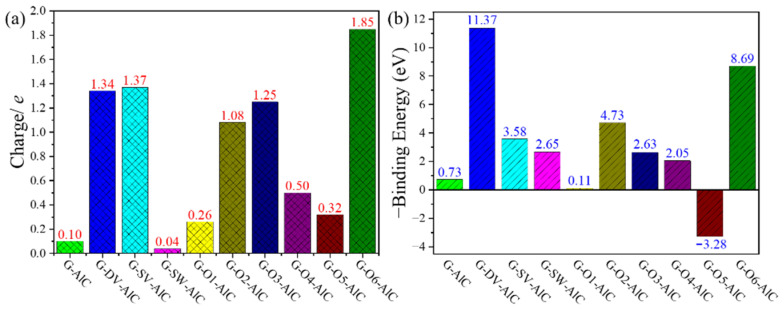
(**a**) Charge transfer amount and (**b**) adsorption binding energy of Al-C cluster adsorption structures for graphene defect models.

**Table 1 materials-15-07312-t001:** The atomic concentration and peak binding energy of XPS spectrum for C 1s, O 1s and Al 2p.

Orbitals	C 1s				O 1s			Al 2p	
Spectra	Al-C	C=C sp2	C-C sp3	C-O	Al-O	C-O	H-O-H	Al^0^	Al^3+^
Atomic (%)	3.06	36.8	23.8	12.98	7.13	12.02	2.89	0.37	0.95
Peak B.E. (eV)	283.3	284.2	284.8	286.0	531.2	532.3	533.5	73.8	74.8

**Table 2 materials-15-07312-t002:** Defect formation energies (eV) before and after the adsorption of Al-O and Al-C clusters for different graphene defects.

Defect	$E_{f}^{G}$	$E_{f}^{\mathrm{Al}-O}$	$\Delta E_{f}^{\mathrm{Al}-O}$	$E_{f}^{\mathrm{Al}-C}$	$\Delta E_{f}^{\mathrm{Al}-C}$
G−DV	8.622	5.312	−3.310	−2.018	−10.640
G−SV	7.238	4.272	−2.966	4.391	−2.847
G−SW	4.764	4.643	−0.121	2.848	−1.916
G−O1	2.383	1.783	−0.600	3.011	0.628
G−O2	0.464	−1.543	−2.007	−3.531	−3.995
G−O3	3.067	1.514	−1.553	1.170	−1.897
G−O4	6.655	6.331	−0.324	5.336	−1.319
G−O5	6.018	5.953	−0.065	10.033	4.015
G−O6	6.855	1.822	−5.033	−1.096	−7.951

## Data Availability

The data presented in this study are available on request from the corresponding authors.

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
