# Peer review of "Influence of Al-O and Al-C Clusters on Defects in Graphene Nanosheets Derived from Coal-Tar Pitch via Al4C3 Precursor"

_materials, 2022, doi:10.3390/ma15207312_

Round 1

Reviewer 1 Report

1. Did the authors perform any convergence testing for the choice of 1×1×1 k-mesh, since it looks too small. 

2. Some comparison of the calculated values with the experimental data (geometric parameters, etc.) would be a good confirmation of the simulation results.

3. Did the authors observe any changes in the geometry of GNs in their calculations? Its relaxation process could impact on the observed results.

Reviewer 2 Report

Dear Authors,

it was a pleasure to review an article "Influence mechanism of Al-O and Al-C clusters on defects in graphene nanosheets derived from coal-tar pitch via Al4C3 precursor ". The manuscript is well prepared but with few mistakes which should be pointed out:

In experiment”

XPS method – there is no information about excitation source monochromated / non-monochromated ( only XPS,K-Alpha) – this information should be included into the next version.

To which line in C1s region the energy scale was calibrated.

One can see that C-C bond is currently (Figure 1b) is below 284eV, which is not correct. If the main C-C peak is not correctly calibrated Al and Oxygen bonds in both O1s and Al2p regions are also not corrected. In such a case why sp2 and sp3 bonds are not separated in C1s region?

The atomic concentration of analyzed sample should be included as a table or in the text.

What kind of software has been used for data deconvolution?

The idea of further parts of manuscript is interesting and well-presented but in few places images (Fig3.b and Fig.5 and Fig.6b should be better visible.

Best regards

Reviewer 3 Report

General comment

Thanks to the authors for their efforts in this study. For further improvement, the authors are advised to look into the following comments.

Abstract

1.      The authors mentioned in line 14, “However, the defects in the prepared GNs have not been well understood.” Does this statement refer to the result of their previous investigation [36], under section 3.1, which reads, “Hence the intensity ratio in our study demonstrates that a few defects are present in the GNs?” If this is so, then, the authors must revise the statement to portray what is obtainable in their current study.

2.      What is the meaning of Al-O and Al-C? The full meaning must be presented before acronyms

Introduction

1.The introduction is concisely written; however, it is shallow, and therefore, more study must be conducted in order to enhance its robustness. Why using DFT and many more must be stated in the introduction.

2. The following papers are recommended for review: 10.5714/CL.2015.16.2.116; 10.1016/j.matpr.2020.03.522; 10.1002/advs.201500101; 10.1016/j.flatc.2020.100211.

Results and discussion

1.      The statement in lines 89-90, must be revised to more scientific inference.

2.   Authors are advised to provide a table containing the XPS spectra analysis results, with proper referencing.

3.      Equation 1 must be properly referenced. If it is without reference, the authors must show it through derivation

4.      In line 113, the authors wrote demotes instead of, denote.

5.      Why is the contribution of the additional O atom energy of oxygen, negative?

6.      The defect created for G-O2 is not shown. Authors are advised to highlight the defects in their figures

7. Figure 2 must be presented with more clarity and visibility

8. Line 135-136 is confusing. Is it the Al-O that creates or forms the defect? Or the authors are investigating the adsorption of Al-O? This must be addressed.

9. Equation 3 must be referenced.

10. What information is to be gained from figure 3b? This is missing from the manuscript

11. Equations 4 & 5 must be referenced

12.  Lines 192-257, are repetition. It would have been better if the authors consider different approaches in order for them to validate their previous results on Al-O.

13.  The results of the experimentation investigated by the XPS spectrum analyzer seem not to have a correlation with the simulation results. Authors must prove this.

14.  The title says, Influence mechanism of Al-O and Al-C…., however, there was no mechanism described by the authors. Authors are advised to modify the title.

Conclusion

The general revision of the manuscript should have effect on the conclusion

Reviewer 4 Report

Peng Lin et al.; reported a manuscript with the title “Influence mechanism of Al-O and Al-C clusters on defects in graphene nanosheets derived from coal-tar pitch via Al4C3 precursor”.

Since all the properties of graphene nanosheets depend on the number of layers, it is surprising that the authors ignored this issue and did not specify what physical properties the synthesized graphene has, such as the number of layers and dimensions. For this reason, I strongly suggest that if the authors are going to perform their calculations based on synthesized graphene, they should perform other characterization such as TEM, HRTEM, XRD, and Raman. Otherwise, these calculations can be done without graphene synthesis, and the graphene synthesis part is just an added part to the article, which has no logical connection.

Round 2

Reviewer 3 Report

Thanks to the authors for addressing the comments. 

Reviewer 4 Report

It can be accepted to publish now.